# LAY-YOUR-SCENE: OPEN-VOCABULARY TEXT TO LAYOUT GENERATION

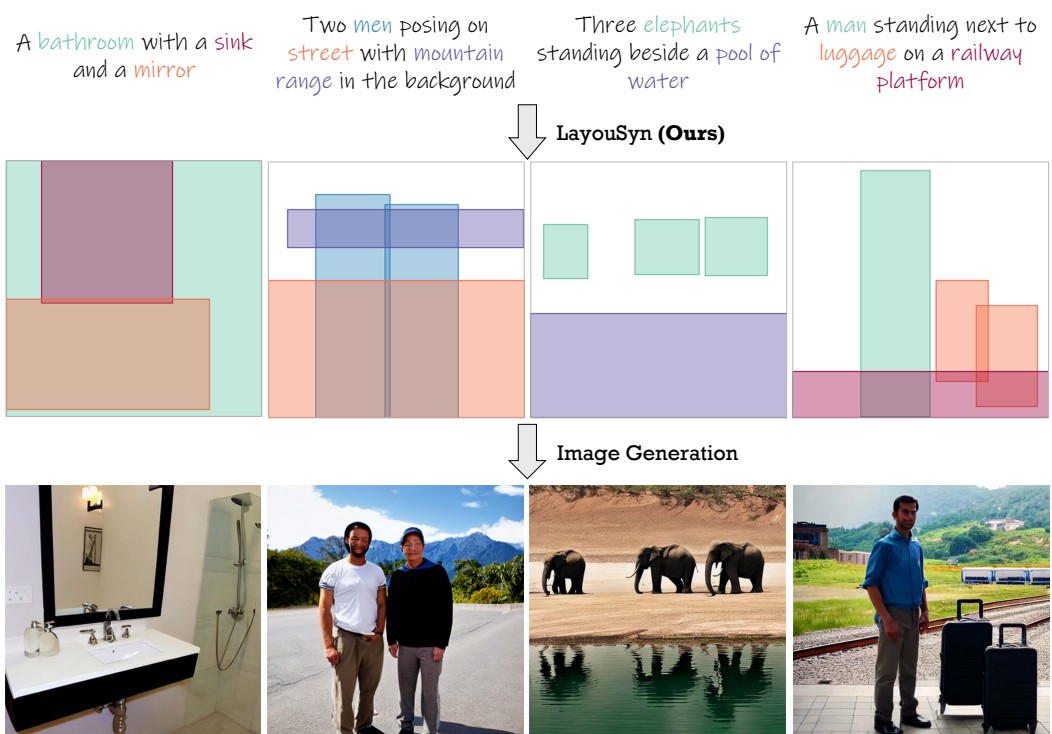

Figure 1: **Open-Vocabulary text to natural scene layout generation** with **LayouSyn** on diverse inputs. LayouSyn demonstrates superior scene awareness compared to existing methods with the ability to generate diverse scene layouts following spatial and numerical constraints.

## ABSTRACT

We present **Lay-Your-Scene** (shorthand *LayouSyn*), a novel diffusion-Transformer based architecture for open-vocabulary natural scene layout generation. Prior works have used close-sourced scene-unaware Large Language models for open-vocabulary layout generation, limiting their widespread use and scene-specific modeling capability. This work presents the first end-to-end text-to-natural-scene-layout generation pipeline that utilizes lightweight open-source language models to predict objects in the scene and a new conditional layout diffusion Transformer trained in a scene-aware manner. Extensive experiments demonstrate that LayouSyn outperforms existing methods on open-vocabulary and closed-vocabulary layout generation and achieves state-of-the-art performance on challenging spatial and numerical reasoning tasks. Additionally, we present two applications of LayouSyn: First, we demonstrate an interesting finding that we can seamlessly combine initialization from the Large Language model to reduce the diffusion sampling steps. Second, we present a new pipeline for adding objects to the image, demonstrating the potential of LayouSyn in image editing applications.

# 1 INTRODUCTION

Generating visual layouts, *i.e.* determining the positions, sizes, and categories of elements, plays an indispensable role in downstream vision tasks such as document analysis (Arroyo et al., 2021) and graphical design (Lee et al., 2020). Recent works tackle layout generation through a continuous (Wang et al., 2024b) or discrete (Gupta et al., 2021; Inoue et al., 2023; Zhang et al., 2023a) diffusion process, where Transformers are often used to model the relationships between elements. Although these methods achieve competitive results across various benchmarks, they primarily focus on **unconditional** layout generation, such as document layouts. Additionally, these models either assume a fixed set of object categories or are incapable of dealing with complex text conditions, which limits their applicability in open-vocabulary settings for natural scenes.

With the advancement of text-to-image generative models (Ramesh et al., 2021; Nichol et al., 2021; Rombach et al., 2022; Chen et al., 2023c; Xue et al., 2024), there has been a growing interest in controllable generation (Li et al., 2023; Zhang et al., 2023b), where users can explicitly control the spatial locations (Xie et al., 2023; Wang et al., 2024a), and counts (Binyamin et al., 2024; Yang et al., 2023) of objects in the generated images. While these frameworks can achieve satisfactory control over image generation, users still need to manually supply fine-grained conditioning inputs, such as plausible scene layouts. A text-to-layout generation framework is therefore needed to reduce the manual effort involved. Some works (Feng et al., 2023; Gani et al., 2024) try to automate this process by generating layouts with close-sourced large language models (LLMs) such as ChatGPT (Ouyang et al., 2022) with in-context prompting. While LLMs can generate reasonable scene layouts, they often produce unrealistic relative object sizes or unnatural bounding box placements (Gani et al., 2024), especially with longer scene descriptions. Additionally, relying on LLMs introduces opacity in the generation process, along with latency and increased costs.

To overcome these limitations, we introduce LayouSyn (Lay-Your-Scene), an open-vocabulary text-to-natural-scene-layout generation framework that combines the strengths of both language models' open-vocabulary capabilities and the strong inductive bias of vision-based models. Our approach divides the scene layout generation task into two stages. In the first stage, a lightweight language model is used to extract a set of labels from the given prompt. In the second stage, we design a new conditional diffusion-transformer network to predict the scene layout, working directly within the bounding box state space.

Our contribution can be summarized as follows:

- **Novel framework and new module**: We propose LayouSyn, the first end-to-end scene-aware text-to-natural-scene-layout generation framework. It adopts small-size language models to predict objects in the scene, and it creates a new conditional diffusion Transformer trained in a scene-aware manner for layout generation. A schematic illustration for the training and inference pipeline can be found in Figure 2.

- **Versatile applications**: We demonstrate the versatility of LayouSyn with two applications. LLM-initialization: we use coarse layouts generated by LLMs such as ChatGPT to initialize LayouSyn, achieving better results with equal or fewer sampling steps. Object-addition: we leverage LayouSyn to perform layout completion, which guides image inpainting to add the new object.

- **State-of-the-art results**: Extensive experiments show that LayouSyn can generate scene layouts that are both semantically and geometrically plausible. LayouSyn outperforms existing methods on multiple closed-vocabulary and open-vocabulary scene generation benchmarks.

# 2 RELATED WORK

**Closed-vocabulary layout generation**   Previous works on closed-vocabulary layout generation focus on a fixed set of object categories and have proposed various architectures to address this task. LayoutGAN (Li et al., 2019), a GAN-based framework, generates both labels and bounding boxes from noise simultaneously. However, it cannot perform generation conditioned on specific label sets, and its evaluations are limited to documents with a small number of elements. LayoutVAE (Jyothi et al., 2019) improves upon this by generating layouts conditioned on label sets autoregressively using LSTM-based VAEs, allowing it to handle a larger number of objects, such as those found in the

COCO dataset (Lin et al., 2015). VTN (Arroyo et al., 2021) further enhances this approach by employing Transformers as the building block for VAEs, better capturing inter-element relationships within a layout. Another line of research formulates layout generation as a sequence generation problem, effectively addressed using Transformers. BLT (Kong et al., 2022) employs a bidirectional Transformer for iterative decoding, while LayoutTransformers (Gupta et al., 2021) uses the standard next-token prediction approach. LayoutFormer++ (Jiang et al., 2023) introduces decoding space restrictions to align layouts more effectively with user-defined constraints. More recently, diffusion models have been explored for layout generation. Dolfin (Wang et al., 2024b) applies continuous diffusion in the bounding box coordinate space, while LayoutDM (Inoue et al., 2023) and LayoutDiffusion (Zhang et al., 2023a) address the task using discrete diffusion on both coordinate and type tokens. Beyond unconditional generation, these models also demonstrate utility in conditional generation tasks, such as layout refinement and type-conditioned generation. Despite these progresses the majority of these works are benchmarked on document layouts, and the closed-vocabulary nature of the models limits their generalizability to layouts for natural scenes.

**Open-vocabulary layout generation**    Open-vocabulary layout generation is an important task that is often coupled with controllable text-to-image generation. For example, GLIGEN (Li et al., 2023), ReCo (Yang et al., 2023), and Boxdiff (Xie et al., 2023) can generate images based on a given scene layout and corresponding region prompts. This requires open-vocabulary layouts, where object categories are not limited to a predefined set, but can include any valid nouns from natural language. Recent approaches predominantly address this challenge by leveraging the reasoning capabilities of large language models (LLMs) like ChatGPT (Ouyang et al., 2022). For instance, LayoutGPT (Feng et al., 2023) introduces a style sheet-like structural language, combined with in-context exemplars to generate layouts with GPT models. Additionally, it proposes Numerical and Spatial Reasoning (NSR-1K) to assess the spatial and counting accuracy in generated layouts, a benchmark we also use to evaluate our LayouSyn. LLM Blueprint (Gani et al., 2024) goes further by generating object descriptions alongside layouts to better guide image generation. While these approaches achieve promising results, their reliance on LLMs reduces transparency and can introduce additional computational costs. In contrast, our LayouSyn relies on a smaller, more efficient language model that can be hosted locally, yet demonstrates strong open-vocabulary capabilities and surpasses competing methods across various benchmarks.

**Diffusion Transformers**    Diffusion Transformers were first introduced in (Peebles & Xie, 2023) to address class-conditional image generation. The self-attention layers in Transformers allow for more effective modeling of relationships between tokens. Beyond text-to-image generation (Esser et al., 2024), this architecture has been adapted for layout generation (Inoue et al., 2023; Wang et al., 2024b), 3D shape generation Mo et al. (2023); Xu et al. (2024), and video generation Brooks et al. (2024). Our approach builds upon Diffusion Transformers for layout generation but operates directly on the continuous bounding box coordinate space, without the need for any VAE encoding.

## 3 METHODOLOGY

This section describes our approach to generating a natural scene layout conditioned on the text prompt and label set in an open-vocabulary manner. Formally, we define a layout $\mathcal{L} = \{(o_i, b_i)\}_{i=1}^{N}$, where $o_i$ is the natural language description or label of the $i^{th}$ object and $b_i \in \mathbb{R}^4$ represents a bounding box in the (top-left, bottom-right) format. Our objective is to generate the layout $\mathcal{L}$ conditioned on the text prompt $p$ and object label set $\mathcal{O}$. We provide a brief overview of diffusion models in Sec. 3.1, describe our architecture in Sec. 3.2, discuss the need for scaling inputs in Sec. 3.3, and present an automated approach to generating label set $\mathcal{O}$ in Sec. 3.4.

### 3.1 PRELIMINARIES

Diffusion models (Ho et al., 2020) are widely used in generative modeling tasks (Ho et al., 2022b;a; Chen et al., 2023a; Cheng et al., 2023; Dhariwal & Nichol, 2021) and are trained to generate samples from a target distribution $p(x)$ by iteratively applying a denoising process to noisy samples, starting from pure Gaussian noise. The forward diffusion process is modeled as a Markov chain, and given a starting sample $x_0 \sim p(x)$, the forward process generates a sequence of samples $\{x_t\}_{t=1}^{T}$ by

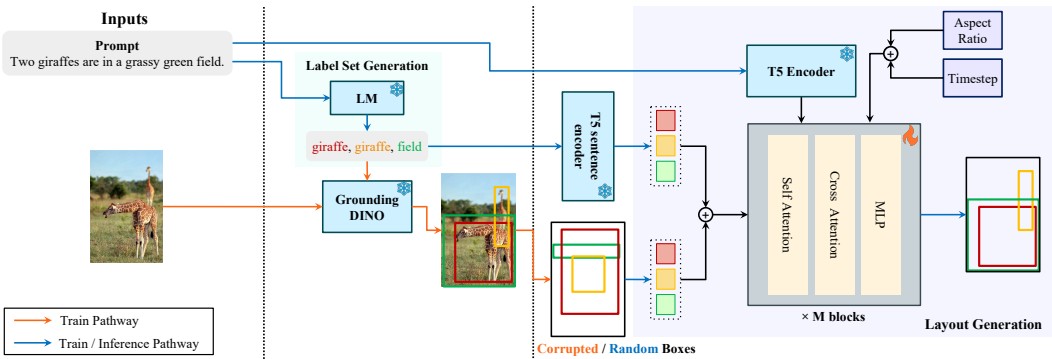

Figure 2: **Overview of Training and Inference pipeline for LayouSyn**: During training, given a supervised (image, caption) pair, we use a lightweight language model to extract label set from the caption and use GroundingDINO (Liu et al., 2023) to extract bounding box coordinates for objects in the label set. Then, we train LayouSyn conditioned on the caption, label set, and aspect ratio of the image. During inference, we generate layouts conditioned on the input prompt, aspect ratio, and label set extracted from the prompt with LM, starting from Gaussian noise.

iteratively adding noise for $T$ timesteps. The forward process is defined as:

$$x_t = \sqrt{\bar{\alpha}_t} \cdot x_0 + \sqrt{1 - \bar{\alpha}_t} \cdot \epsilon_t, \ \epsilon_t \sim \mathcal{N}(0, I) \tag{1}$$

where $\bar{\alpha}_t$ is the noise schedule, which decreases from 1 to 0 as $t$ goes from 0 to $T$ in the diffusion process. A denoiser $\epsilon_\theta$ is trained to predict the noise added to the sample $x_0$ at a given timestep $t$. The denoiser is modeled as a neural network with parameters $\theta$ and is trained to minimize the MSE loss between added noise $\epsilon_t$ and the predicted noise $\epsilon_\theta(x_t, t)$:

$$\mathcal{L}(\theta) = \mathbb{E}_{x_0 \sim p(x), t \sim \mathcal{U}(1,T)} \left[ \|\epsilon_\theta(x_t, t) - \epsilon_t\|^2 \right] \tag{2}$$

## 3.2 ARCHITECTURE

We adopt Diffusion-Transformer (DiT) architecture for denoising and operate directly in bounding box coordinate space to generate layouts conditioned on the text prompt $p$, object label set $\mathcal{O}$, and aspect ratio ar. We scale the bounding box coordinates by width and height to range $[0, 1]$ and further normalize the coordinates to the range $[-1, 1]$. We encode the object label $o_i$, bounding box coordinates $b_i$, and position $i$ of an object $o_i$ into a single fixed-size $d$-dimensional token $t_i \in \mathbb{R}^d$. Formally, the token is computed as:

$$t_i = \texttt{MLP}(b_i) + \texttt{Embedder}(o_i) + \texttt{PositionalEncoding}(i) \tag{3}$$

where `Embedder` is a sentence embedding model that maps the object label $o_i$ to a fixed-size embedding, `MLP` is a multi-layer perceptron that maps the bounding box coordinates to $d$-dimensional embedding, and `PositionalEncoding` is 1D sinusoidal positional encoding. We condition the denoiser on the timestep $t$ and the aspect ratio ar $= L_w/L_h$, where $L_w$ and $L_h$ are the width and height of the layout, respectively and incorporate the global conditioning information with adaptive layer normalization (Perez et al., 2017). Finally, we modify the DiT blocks and add a cross-attention layer (Chen et al., 2023b) to incorporate information from the text prompt $p$. We visualize the complete architecture in Fig. 2.

## 3.3 SCALING

The signal-to-noise ratio significantly affects the performance of the diffusion model (Chen, 2023), and the low dimensionality of bounding box coordinates results in information being destroyed in the initial phases of the denoising process, as demonstrated in Appendix Fig. A.2. Previous works (Chen et al., 2023e;d) have proposed to scale the input to the denoiser by a scaling factor $s$. However, this approach requires normalization of inputs for a stable training (Chen, 2023). Instead, we propose to

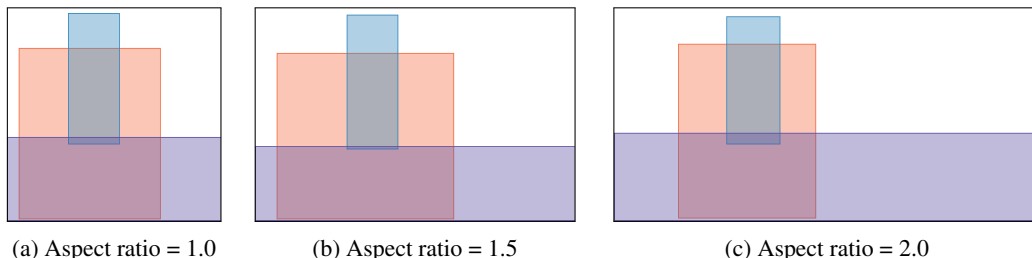

(a) Aspect ratio = 1.0    (b) Aspect ratio = 1.5    (c) Aspect ratio = 2.0

Figure 3: **Layout Generation with varying aspect ratio**: Layouts generated at different aspect ratios for prompt: *A man riding a horse on the street*. The model adjusts the position and aspect ratio of the bounding box corresponding to the man and the horse to produce natural looking layouts.

incorporate the scaling factor directly in the noise schedule $\alpha_t$:

$$\bar{\alpha}'_t = \frac{\bar{\alpha}_t \cdot s^2}{1 + (\bar{\alpha}_t \cdot (s^2 - 1))} \tag{4}$$

We visualize the effect of the scaling factor on the denoising process in Appendix Fig. A.2 and provide complete proof in Theorem 1. Overall, $s > 1$ results in a more gradual destruction of information for the bounding box coordinates and improves the performance of the diffusion model, as demonstrated in our ablation study.

### 3.4 LABEL SET GENERATION

The label set $\mathcal{O}$ is a function of prompt $p$ and contains the object labels present in the scene described by the prompt. A large language model (LLM) trained on a large corpus of text data is suitable to predict the object labels and their counts from the prompt. We prompt LLM to extract noun phrases from the prompt, assign a count to each noun phrase, and filter out the noun phrases that cannot be visualized in the scene. This allows us to generate the label set $\mathcal{O}$ from the prompt $p$ in an open-vocabulary manner, and our method can work in settings where object labels are not present or provided by the users. The details for prompting LLM are in Appendix A.2 and we visualize a few results in Tab. 1.

Table 1: Examples of label sets generated with LLama3.1-8B.

| Prompt | Label set |
|---|---|
| There is a teapot and food on a plate. | teapot: 1, food: 1, plate: 1 |
| Two men carrying plastic containers walking barefoot in the sand. | man: 2, plastic container: 2, sand: 1 |
| A couple of children sitting down next to a laptop computer. | child: 2, laptop: 1 |
| A man riding on the back of an elephant along a dirt road. | man: 1, elephant: 1, dirt road: 1 |
| Girl on a couch with her computer on a table | girl: 1, couch: 1, computer: 1, table: 1 |

## 4 EXPERIMENTS

We conduct a comprehensive set of experiments to demonstrate that LayouSyn achieves state-of-the-art performance on both closed-vocabulary and open-vocabulary scene layout generation in Sec. 4.1 and Sec. 4.2 respectively. Additionally, we conduct ablation studies in Sec. 4.3 to demonstrate the effectiveness of our choices pertaining to scaling factor and use of LLMs for label set generation.

### 4.1 CLOSED-VOCABULARY LAYOUT GENERATION

Closed-vocabulary layout generation is the task of generating a layout $\mathcal{L}$ conditioned on the label set $\mathcal{C} = \{c_1, c_2, \ldots, c_n\}$, where each object label $c_i$ is from a fixed vocabulary $\mathcal{V}$.

Table 2: **Closed-Vocabulary Evaluation on COCO17 dataset**: LayouSyn-CV achieves state-of-the-art performance on FID metrics and comparable performance on the IS metric.

| Model | Layout Eval. | Image Eval. | |
|---|---|---|---|
| | FID ↓ | IS ↑ | FID ↓ |
| GroundTruth | 0.0 | 7.82 | 80.21 |
| LayoutVAE | 5.23 | 7.14 | 80.65 |
| LayoutTransformer | 18.47 | **7.76** | 81.63 |
| LayouSyn-CV | **4.78** | 7.21 | **79.98** |

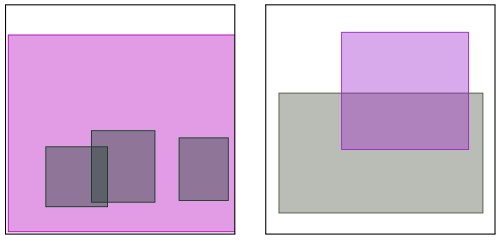

(a) Labels: bed, teddy bear    (b) Labels: cat , bowl

Figure 4: **Layouts generated by LayouSyn-CV**: Our model demonstrates the ability to understand spatial relationships and aspect ratios of objects in the scene.

**Setup**    Following previous works (Jyothi et al., 2019; Feng et al., 2023), we evaluate LayouSyn on COCO17 (Lin et al., 2015) Instance dataset. We modify LayouSyn to remove cross-attention layers and use 6 DiT blocks, each with 6 heads for multi-head attention, and a hidden dimension of size 144. We use 250 diffusion steps at a scale of 2.0, Adam (Kingma & Ba, 2017) optimizer with a learning rate of $10^{-4}$, batch size 32, and train for 1M steps on 1 NVIDIA RTX A5000 GPUs. We sample with 250 DDPM steps. We call this architecture **LayouSyn-CV**.

**Baselines**    We compare our work with LayoutVAE and LayoutTransformer trained on the COCO17 Instance dataset using the open-source implementation provided by LayoutTransformer. Note that LayouSyn and LayoutVAE are conditioned on the label set, whereas LayoutTransformer predicts the next object labels during generation. To ensure a fair comparison, we modify the sampling algorithm of LayoutTransformer and force the token predictions to match the label set.

**Metrics**    We evaluate the quality of generated layouts on two criteria: (1) **Layout Quality:** Following Document Layout Generation literature (Wang et al., 2024b; Jyothi et al., 2019; Li et al., 2019), we draw the layout as an image and map each object to a specific color and compare the generated images using Fréchet Inception Distance (FID) (Heusel et al., 2018). (2) **Image Quality:** We use Layout2Im (Zhao et al., 2019) to generate images from layouts and compute the Fréchet Inception Distance (FID) (Heusel et al., 2018) and Inception Score (IS) (Salimans et al., 2016) with the COCO17 Instance validation dataset.

The results are reported in Tab. 2, and we visualize layouts generated by LayouSyn-CV in Fig. 4. LayouSyn-CV achieves state-of-the-art performance on the FID metric and comparable performance on the IS metric. We believe our method achieves better results due to two reasons: (1) We operate directly in the bounding box space, unlike LayoutVAE, which operates in the latent space and leads to loss of information during the encoding-decoding process, and (2) We handle the object labels more straightforwardly by simply adding embedding to the input tokens. Overall, our results demonstrate the effectiveness of LayouSyn for closed-vocabulary layout generation.

## 4.2 OPEN VOCABULARY LAYOUT GENERATION

Open-vocabulary layout generation is the task of generating layout $\mathcal{L}$ conditioned on the prompt $p$ where the object labels and prompts can be any sentence in the natural language.

**Training**    We use LLama3.1-8B model for predicting the label set $\mathcal{C}$ from the text prompt $p$ and LayouSyn diffusion Transformer for generating the layout conditioned on the label set $\mathcal{C}$ and the prompt $p$. LayouSyn architecture consists of 6 DiT blocks, each with 4 attention heads for multi-head attention, and a hidden dimension of size 256, resulting in a denoiser with ∼10M parameters. We use 1000 diffusion steps at a scale factor of $s = 2.0$ for training and sample with 250 DDPM steps. We use Adam (Kingma & Ba, 2017) with a learning rate of $10^{-4}$, batch size 32, and train for 725K steps on 2 NVIDIA RTX A5000 GPUs. We have described the datasets used for training LayouSyn below:

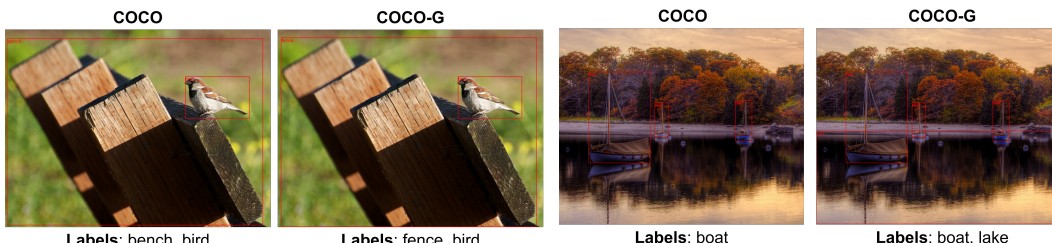

(a) Incorrect object label          (b) Missing object annotation

Figure 5: We visualize bounding boxes present in the COCO dataset and COCOGroundedDataset. For each image, the left side shows the original image from the COCO dataset, while the right side shows the corresponding image with the bounding boxes from the COCOGroundedDataset.

Table 3: **Open-Vocabulary Layout Quality Evaluationon COCO**: FID scores for layout quality evaluation on COCO.

| Model | FID ↓ |
|---|---|
| LayoutGPT (GPT-3.5-chat) | 5.02 |
| LayouSyn | **3.54** |

Table 4: **Open-Vocabulary Human Evaluation on COCO**: Mean score and standard deviation for Layout quality rating on the scale of 1-5.

| Model | Mean Score ↑ |
|---|---|
| LayoutGPT-3.5-chat | 3.53 (± 1.26) |
| LayoutGPT-4 | 3.75 (± 1.14) |
| LayouSyn | **3.89** (± 1.12) |

1. **NSR-1K Spatial:** We use the NSR-1K spatial dataset proposed in LayoutGPT (Feng et al., 2023) to train our model for understanding spatial relationship between objects present in the scene. The dataset contains 738 prompts describing four spatial relations: *above*, *below*, *left*, and *right* between two objects in the scene.

2. **COCOGroundedDataset (COCO-GR):** COCO17 (Lin et al., 2015) is a widely known dataset containing image-caption pairs along with bounding boxes of objects present in the image. However, there are two limitations with directly using the COCO17 dataset for training LayouSyn: (1) The labels of bounding boxes are limited to 80 object classes, limiting the ability to train an open-vocabulary model, and (2) There is a low semantic overlap between the bounding boxes of objects in the image and the associated captions as visualized in Fig. 5. To address these issues, we create a *Grounded MS-COCO dataset* following (Peng et al., 2023), which we refer to as **COCO-GR**. We extract nouns present in the image captions with LLama and obtain the bounding boxes for the extracted nouns using GroundingDINO. Our dataset generation pipeline is visualized in Fig. 2. Our final dataset contains 578,951 layouts with an average of 5.62 objects per layout and an average prompt length of 9.91 words.

### 4.2.1 COCO EVALUATION

We evaluate LayouSyn on the COCO17 validation dataset and compare the performance with LayoutGPT (Feng et al., 2023), which, to the best of our knowledge, is the only work for open-vocabulary natural scene layout generation. For a fair comparison, we add the COCO-GR training dataset to the in-context exemplars used by LayoutGPT. The generated layouts are evaluated on two criteria: **Layout Quality:** We draw the layout as an image and map each object to a specific color, taking into account semantic similarity between different objects based on CLIP (Radford et al., 2021) similarity, and compare the generated images using Fréchet Inception Distance (FID) (Heusel et al., 2018) with the *COCO-GR* validation dataset. Due to cost constraints with using GPT models, we limit our evaluation to the first 8700 captions from the COCO validation dataset and use

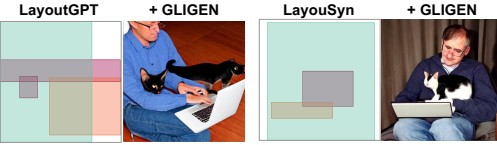

**Caption**: A man on a snowboard para sailing in the snow.    **Caption**: A cat sitting on the man's lap while the man types on the laptop.

Figure 6: Qualitatively comparing LayouSyn with LayoutGPT. **Top**: LayouSyn generates label sets strictly following the caption; **bottom**: LayouSyn can generate complex layouts with multiple objects following spatial constraints in the prompt.

Table 5: **Spatial and Counting evaluation on NSR-1K benchmark**: LayouSyn outperforms existing methods on spatial and counting reasoning tasks and achieves state-of-the-art performance on most metrics. Note: '*' denotes metric reported by LayoutGPT Feng et al. (2023).

| | Numerical Reasoning | | | | | Spatial Reasoning | | |
|---|---|---|---|---|---|---|---|---|
| | Prec. ↑ | Recall ↑ | Acc. ↑ | GLIP ↑ | CLIP ↑ | Acc. ↑ | GLIP ↑ | CLIP ↑ |
| GT layouts | 100.0 | 100.0 | 100.0 | 50.08 | 0.258 | 100.00 | 57.20 | 0.259 |
| LayoutTransformer* | 75.70 | 61.69 | 22.26 | 40.55 | 0.247 | 6.36 | 28.13 | 0.241 |
| LayoutGPT (LLama2-13B) | 78.92 | 83.41 | 68.06 | 44.78 | 0.259 | 45.02 | 28.90 | 0.261 |
| LayoutGPT (LLama3-8B-Instruct) | 78.61 | 84.01 | 71.71 | 49.25 | 0.261 | 75.41 | 47.49 | 0.263 |
| LayoutGPT (GPT-3.5-Chat) | 76.29 | 86.64 | 76.72 | 54.25 | **0.263** | 87.07 | 56.89 | **0.266** |
| LayoutGPT (GPT-4) | **81.02** | 85.63 | 78.11 | 52.02 | 0.260 | 91.59 | 58.02 | **0.266** |
| **LayouSyn(Ours)** | 77.62 | **99.23** | **95.14** | **55.54** | 0.262 | **92.15** | **59.29** | 0.265 |

LayoutGPT with GPT-3.5. **Human Evaluation:** We perform a human evaluation on 100 randomly selected captions from the COCOCaptioning validation dataset for LayouSyn and LayoutGPT with GPT-3.5 and GPT-4. Our survey was completed by graduate students with an average of 4.6 ratings per layout, and we plan to further conduct a larger-scale AMT study. More details on the human evaluation setup are provided in Appendix Appendix A.3.

We visualize generated layouts and corresponding images generated with GLIGEN (Li et al., 2023) in Fig. 6 and the results for Layout Quality evaluation and Human evaluation are shown in Tab. 3 and Tab. 4 respectively. We outperform LayoutGPT on both FID by 29.48% and achieve a better average rating by 0.14 points on the human evaluation, demonstrating the superiority of LayouSyn in open-vocabulary layout generation.

### 4.2.2 SPATIAL AND NUMERICAL EVALUATION

We evaluate LayouSyn on the NSR-1K spatial and numerical reasoning benchmark and compare our results with LayoutGPT (Feng et al., 2023). We use GLIGEN (Li et al., 2023) to generate images from layouts and, for a fair comparison, re-run GLIGEN on the layouts reported in LayoutGPT due to lack of original hyperparameters. We briefly describe the metrics below for completeness and refer the readers to LayoutGPT (Feng et al., 2023) for more details.

1. **Numerical Reasoning:** We evaluate the numerical quality of the generated layouts on Precision, Recall, Accuracy, GLIP accuracy, and CLIP similarity. *Precision* is the percentage of predicted objects in the ground-truth objects set, and *Recall* is the percentage of ground-truth objects in the predicted object set. *Accuracy* for a test example is defined as 1 if the ground-truth object set and predicted object set overlap exactly and 0 otherwise. The *GLIP accuracy* for a test example is defined as 1 if the GLIP detected object count matches the ground-truth object count and 0 otherwise. The *CLIP similarity* is the cosine similarity between the CLIP embeddings of the generated image and the input prompt features.

2. **Spatial Reasoning:** We evaluate spatial reasoning on accuracy, GLIP accuracy, and CLIP similarity. *Accuracy* and *GLIP accuracy* for a test example is defined as 1 if the predicted object locations in layout and GLIP detected bounding box follow the spatial constraints, and 0 otherwise. *CLIP similarity* is defined in the same as numerical reasoning.

Table 6: Model evaluation for LayouSyn with different scales and configurations and DDPM sampling with 250 steps.

| Scale | CFG | Acc. ↑ | GLIP ↑ | CLIP ↑ | FID ↓ |
|---|---|---|---|---|---|
| | 1.0 | 89.32 | **57.88** | 0.266 | 3.5 |
| 1 | 2.0 | 90.24 | 57.1 | **0.267** | 3.2 |
| | 4.0 | **91.87** | 55.62 | 0.266 | **2.97** |
| | 1.0 | 88.62 | **58.45** | **0.267** | 3.29 |
| 2 | 2.0 | **92.36** | 59.29 | 0.265 | **3.08** |
| | 4.0 | **93.0** | 57.6 | 0.266 | **2.97** |
| | 1.0 | 88.40 | **59.01** | 0.266 | 3.18 |
| 3 | 2.0 | 90.95 | 58.02 | 0.265 | 3.04 |
| | 4.0 | 91.02 | **58.09** | 0.266 | **3.01** |

Table 7: Effect of LLMs on the object label set generation

| Model | FID ↓ |
|---|---|
| LLama-3.2-3B-Instruct | 4.72 |
| LLama-3.1-8B-Instruct | **3.08** |
| GPT-3.5-chat | 3.97 |

Table 8: Spatial reasoning results with LLM initialization. Label Set: using the label set derived from LLM; Inv: initialize bounding boxes with DDIM inversion of LLM predictions (numbers in bracket are steps of inversion performed)

| | Acc. ↑ | GLIP ↑ | CLIP ↑ |
|---|---|---|---|
| LLama2-13B | 45.02 | 28.90 | 0.261 |
| Label Set | 88.90 | **58.23** | **0.265** |
| Label Set + Inv (150) | **89.33** | 57.31 | **0.265** |
| LLama3-8B-Instruct | 75.41 | 47.49 | 0.263 |
| Label Set | 87.70 | 57.10 | 0.264 |
| Label Set + Inv (150) | **89.05** | **58.73** | **0.265** |
| GPT-3.5-Chat | 87.07 | 56.89 | 0.266 |
| Label Set | 89.54 | 57.95 | 0.266 |
| Label Set + Inv (250) | **90.11** | **58.30** | 0.266 |
| GPT-4 | **91.59** | 58.02 | 0.266 |
| Label Set | 91.17 | 58.52 | 0.265 |
| Label Set + Inv (100) | **91.59** | **60.14** | 0.265 |

The results on the NSR-1K benchmark are reported in Tab. 5. LayouSyn achieves superior performance across multiple metrics, including 92.15% accuracy in spatial reasoning, 59.29% GLIP detection accuracy, and a recall of 99.23% and an accuracy of 95.14%. Note that a high recall indicates a very high overlap between the predicted and ground-truth objects set, indicating that smaller language models can be effectively used for object label generation.

### 4.3 ABLATION STUDY

**Scale**  We report the results on ablation with different scales and CFG scales in Tab. 6. We observe that the model trained with scale 2.0 achieves the best performance on all metrics. A scale of 2.0 with CFG 2.0 achieves the best performance on most metrics. Overall, we observe that the performance first increases with scale and then decreases. We believe that the decrease in the performance with higher scales is due to the noise schedule dropping too quickly in the later diffusion steps (Appendix Fig. A.2).

**Object label generation techniques**  We evaluate the performance of LayouSyn with label set generated with LLama3.2-3B-Instruct, LLama3.1-8B-Instruct, and GPT-3.5-chat to study the effect of model size on the quality of generated layouts. The results are reported in Tab. 7 and our method achieves the best performance with LLama3.1-8B-Instruct. These results strengthen our claim that smaller language models can be effectively used for object label generation since parsing the object labels from a prompt is a simpler task compared to generating the layouts with the language models.

### 5 APPLICATIONS

#### 5.1 LLM INITIALIZATION

LayouSyn can be integrated with an LLM, using its planned layouts as initialization and refining them to achieve better performance with equal or fewer sampling steps. Specifically, we take the outputs from LayoutGPT, which can be used with different LLMs. For initialization, we design two strategies: 1) *Label set only*: use only the label sets $\mathcal{O}$ predicted by the LLM and perform denoising starting from Gaussian noise. Full 250 denoising steps are executed; 2) *Label Set + Inversion*: in addition to using the label sets, apply DDIM inversion (Couairon et al., 2022) on the bounding boxes predicted by the LLM. We only denoise for the same number of steps as inversion.

We present spatial reasoning evaluations in Tab. 8. When using only label sets, LayouSyn brings a large improvement in accuracy for Llama2 (+43.88) and LLama3 (+12.29), and outperforms GPT-3.5 by 2.47. Comparing the results from Gaussian noise initialization (Label Set) with those from

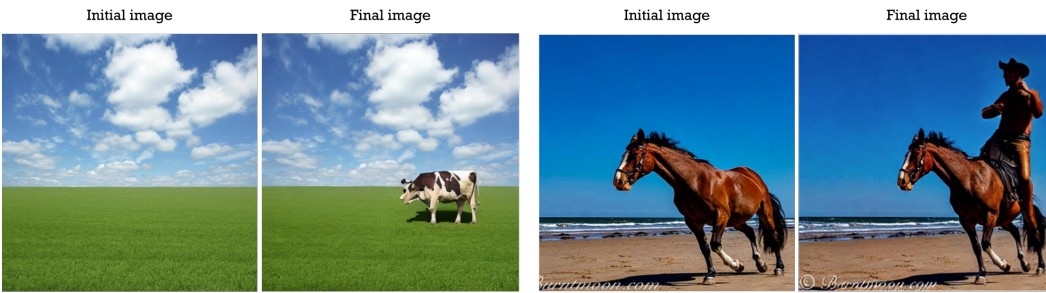

**Prompt**: A cow grazing in the field below blue sky
**Object to add**: cow

**Prompt**: A man riding a horse on the beach
**Object to add**: man

Figure 7: **Automated object addition using LayouSyn**: Our pipeline consists of four steps: extracting relevant objects from the prompt with LLM, detecting objects present in the scene with GroundingDINO (Liu et al., 2023), layout completion of the object to add with LayouSyn (ours), and finally inpainting the object in the image with GLIGEN (Li et al., 2023).

DDIM inversion (Label Set + Inv), the latter consistently yields higher accuracy, often requiring fewer than 250 sampling steps, regardless of the LLM used. This highlights the effectiveness of LLM initialization compared to pure Gaussian noise, even when the LLM predictions are coarse.

## 5.2 Object addition Pipeline

Image inpainting (Lugmayr et al., 2022) with the diffusion model is widely used for adding objects to images. However, these models need users to specify the spatial location of the objects to be added, requiring a human-in-the-loop to guide the inpainting process. In this paper, we answer the following question: Given an image $I$, a list of objects to add to the image $\mathcal{A}$, and a prompt $p$ describing the final image, can we add objects to the image without human intervention in an automated manner? To the best of our knowledge, this is the first work that addresses the problem of adding objects to images without any human intervention in an end-to-end pipeline with layout completion. We discuss components of our pipeline below and visualize examples in Fig. 7.

1. **Label set:** We use an LLM to generate an object set $\mathcal{O}$ from the prompt $p$. $\mathcal{O}$ contains a list of objects that need to be considered during the object addition process.

2. **Object Detection:** We use a pre-trained object detection model to detect objects from the label set $\mathcal{O}$ in the image $I$. We obtain a set of bounding boxes $\mathcal{B}$ for the detected objects and create a layout $L$ with obtained bounding boxes $\mathcal{B}$ and label set $\mathcal{O}$.

3. **Layout Completion:** We inpaint (Lugmayr et al., 2022) the bounding box locations of the objects to add with LayouSyn, and obtain an inpainting mask $M$ based on the predicted bounding boxes for objects in $\mathcal{A}$.

4. **Object Inpainting:** We use inpainting (Lugmayr et al., 2022) with GLIGEN (Li et al., 2023) to inpaint objects in the set $\mathcal{A}$ into the image $I$ using the inpainting mask $M$.

## 6 Conclusion

We present **Lay-Your-Scene** (abbreviated as *LayouSyn*), a novel diffusion Transformer architecture for open-vocabulary natural scene layout generation. We demonstrate that LayouSyn can be combined with small-sized LLMs for an end-to-end text-to-layout generation pipeline. Extensive experiments demonstrate that LayouSyn outperforms existing methods on multiple layout generation benchmarks, including the challenging spatial and numerical reasoning tasks. Further, we demonstrate an interesting finding that we can seamlessly combine initialization from LLMs to reduce the diffusion sampling steps and refine the LLM predictions. Finally, we present a new pipeline for adding objects to the image, demonstrating the potential of LayouSyn in image editing applications.

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

## A   APPENDIX

### A.1   SCALING FACTOR

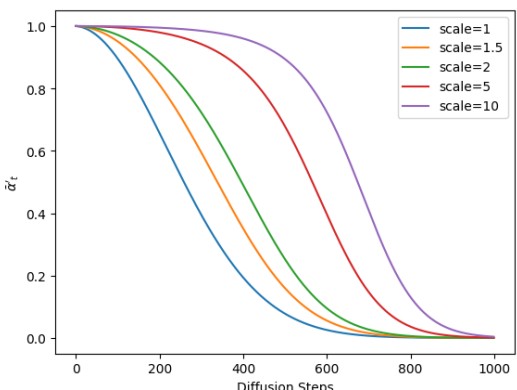

Figure A.1: **Effect of Scaling Factor on Denoising Process**: We plot the noise schedule $\bar{\alpha}'_t$ for diffusion process with 1000 steps for different scaling factors $s$. We observe that $s > 1$ results in a more gradual destruction of information.

**Theorem 1.** *Given the forward process scaled by a factor s:*

$$X_t = s\sqrt{\alpha_t}X_0 + \sqrt{1 - \alpha_t}\epsilon_t, \quad \epsilon_t \sim \mathcal{N}(0,1) \tag{A.1}$$

*with the assumptions*

$$\mathbb{E}[X_0] = 0 \quad and \quad Var(X_0) = 1, \tag{A.2}$$

*the normalized process $\tilde{X}_t$ given by*

$$\tilde{X}_t = \frac{\sqrt{\alpha_t}sX_0 + \sqrt{1 - \alpha_t}\epsilon_t}{\sqrt{(s^2 - 1)\alpha_t + 1}} \tag{A.3}$$

*has the property that $Var(\tilde{X}_t) = 1$, and the corresponding coefficient $\tilde{\alpha}_t$ for $X_0$ is*

$$\tilde{\alpha}_t = \frac{\sqrt{\alpha_t}s}{\sqrt{(s^2 - 1)\alpha_t + 1}}. \tag{A.4}$$

*Proof.* We start with the expression for $X_t$:

$$X_t = s\sqrt{\alpha_t}X_0 + \sqrt{1 - \alpha_t}\epsilon_t. \tag{A.5}$$

**Step 1: Expectation of $X_t$**
Taking the expectation of both sides:

$$\mathbb{E}[X_t] = \mathbb{E}\left[s\sqrt{\alpha_t}X_0 + \sqrt{1 - \alpha_t}\epsilon_t\right]. \tag{A.6}$$

Since $\mathbb{E}[X_0] = 0$ and $\mathbb{E}[\epsilon_t] = 0$, it follows that:

$$\mathbb{E}[X_t] = s\sqrt{\alpha_t} \cdot \mathbb{E}[X_0] + \sqrt{1 - \alpha_t} \cdot \mathbb{E}[\epsilon_t] = 0. \tag{A.7}$$

Thus,

$$\mathbb{E}[X_t] = 0. \tag{A.8}$$

**Step 2: Variance of $X_t$**
Next, we compute the variance of $X_t$:

$$\mathrm{Var}(X_t) = \mathbb{E}[X_t^2] - \mathbb{E}[X_t]^2 = \mathbb{E}[X_t^2]. \tag{A.9}$$

Since $\mathbb{E}[X_t] = 0$, we simplify $\mathrm{Var}(X_t)$ by expanding $X_t^2$:

$$X_t^2 = \left(s\sqrt{\alpha_t}X_0 + \sqrt{1 - \alpha_t}\epsilon_t\right)^2. \tag{A.10}$$

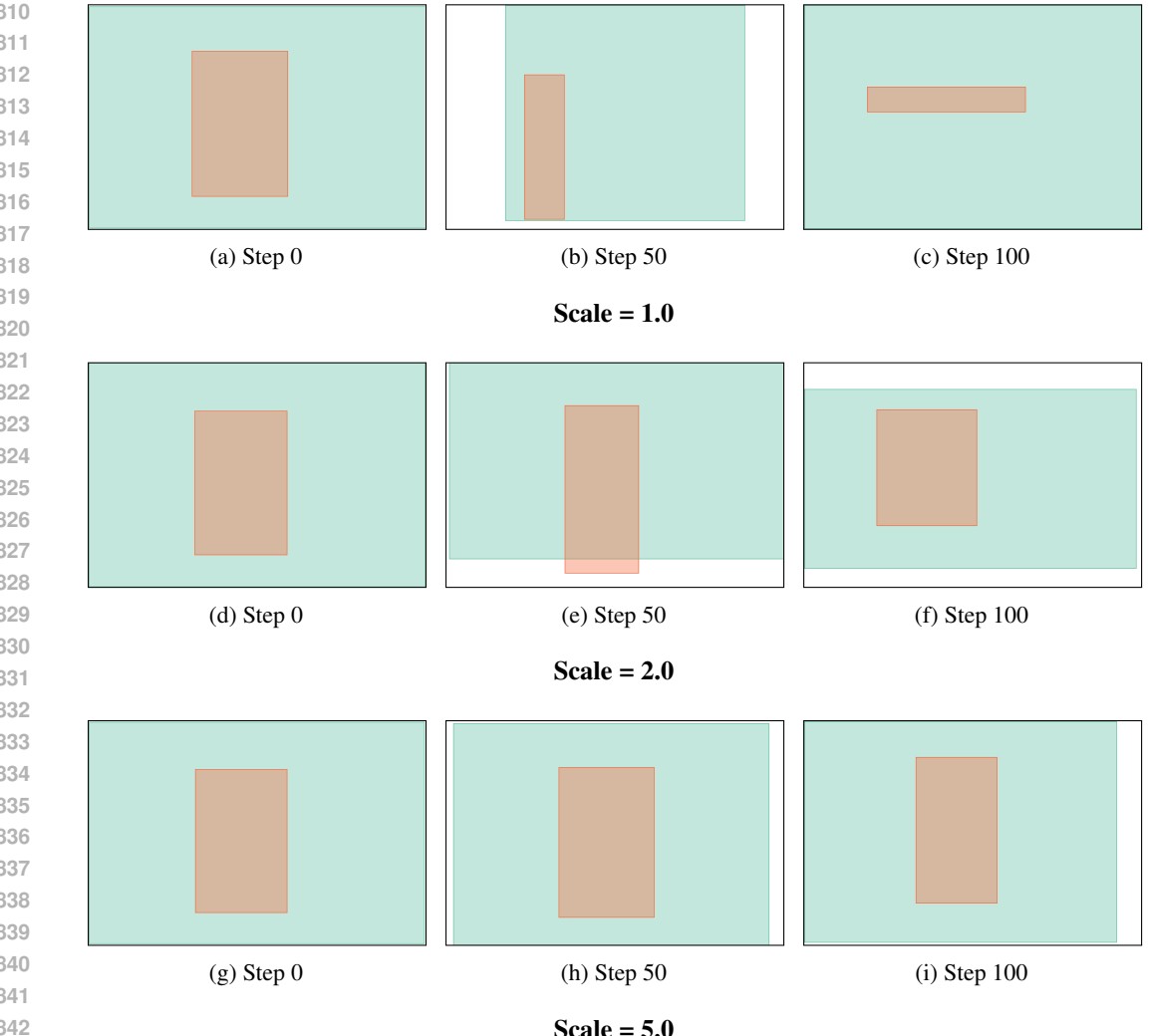

(a) Step 0      (b) Step 50      (c) Step 100

**Scale = 1.0**

(d) Step 0      (e) Step 50      (f) Step 100

**Scale = 2.0**

(g) Step 0      (h) Step 50      (i) Step 100

**Scale = 5.0**

Figure A.2: **Visualizing denoising process scale 1.0, 2.0, and 5.0**: The denoising process for higher scaling factor results in a more gradual destruction of information for the bounding box coordinates for Layout with Prompt: Snowboarder cuts his way down a ski slope

Expanding the square:

$$X_t^2 = s^2\alpha_t X_0^2 + 2s\sqrt{\alpha_t(1-\alpha_t)}X_0\epsilon_t + (1-\alpha_t)\epsilon_t^2. \qquad \text{(A.11)}$$

Taking the expectation:

$$\mathbb{E}[X_t^2] = s^2\alpha_t\mathbb{E}[X_0^2] + 2s\sqrt{\alpha_t(1-\alpha_t)}\mathbb{E}[X_0\epsilon_t] + (1-\alpha_t)\mathbb{E}[\epsilon_t^2]. \qquad \text{(A.12)}$$

Since $\mathbb{E}[X_0^2] = \text{Var}(X_0) = 1$, $\mathbb{E}[\epsilon_t^2] = 1$, and $\mathbb{E}[X_0\epsilon_t] = 0$ (as $X_0$ and $\epsilon_t$ are independent), this simplifies to:

$$\mathbb{E}[X_t^2] = s^2\alpha_t + (1-\alpha_t). \qquad \text{(A.13)}$$

Therefore, the variance of $X_t$ is:

$$\text{Var}(X_t) = s^2\alpha_t + (1-\alpha_t). \qquad \text{(A.14)}$$

This can be rewritten as:

$$\text{Var}(X_t) = \alpha_t(s^2 - 1) + 1. \qquad \text{(A.15)}$$

**Step 3: Normalization of $X_t$**

We define the normalized process $\tilde{X}_t$ as:

$$\tilde{X}_t = \frac{s\sqrt{\alpha_t}X_0 + \sqrt{1 - \alpha_t}\epsilon_t}{\sqrt{(s^2 - 1)\alpha_t + 1}}. \tag{A.16}$$

This normalization ensures that the variance of $\tilde{X}_t$ is 1:

$$\text{Var}(\tilde{X}_t) = \frac{\text{Var}(X_t)}{(s^2 - 1)\alpha_t + 1} = 1. \tag{A.17}$$

**Step 4: Expression for $\tilde{\alpha}_t$**

From the normalized process $\tilde{X}_t$, the corresponding coefficient $\tilde{\alpha}_t$ for $X_0$ is given by:

$$\tilde{\alpha}_t = \frac{\sqrt{\alpha_t}s}{\sqrt{(s^2 - 1)\alpha_t + 1}}. \tag{A.18}$$

This completes the proof. $\qquad\square$

### A.2 LABEL SET GENERATION

We use a pre-trained LLM to generate a set of object labels $\mathcal{O}$ from a given prompt. Our overall is to ask LLM to follow a series of steps to extract the noun phrases from the prompt which can be visualized in the scene and output the object and the count in a JSON format. Our prompt is as follows:

```
You are a creative scene designer who predicts a scene from a natural
    language prompt. A scene is a JSON object containing a list of
    noun phrases with their counts {"phrase1": count1, "phrase2":
    count2, ...}. The noun phrases contain **ONLY** common nouns. You
    strictly follow the below process for predicting plausible
    scenes:

Step 1: Extract noun phrases from the prompt. For example, "happy
    people", "car engine", "brown dog", "parking lot", etc.
Step 2: Limit noun phrases to common nouns and convert the noun
    phrase to its singular form. For example, "happy people" to "
    person", "tall women" to "woman", "group of old people" to "
    person", "children" to "child", "brown dog" to "dog", "parking
    lot" remains "parking lot", etc.
Step 3: Predict the count of each noun phrase and ensure consistency
    with the count of other objects in the scene. If a particular
    object does not have any explicit count mentioned in the prompt,
    use your creativity to assign a count to make the overall scene
    plausible but not too cluttered. For example, if the prompt is "a
     group of young kids playing with their dogs," the count of "kid"
     can be 3, and the count of "dog" should be the same as the count
     of "kid".
Step 4: Output the final scene as a JSON object, only including
    physical objects and phrases without referring to actions or
    activities.

Complete example:

Prompt: Three white sheep and few women walking down a town road.
Steps:
Step 1: noun phrases: white sheep, women, town road
Step 2: noun phrase in singular form: sheep, woman, town road
Step 3: Since the count of women is not mentioned, we will assign a
    count of 2 to make the scene plausible. The count of "sheep" is 3
     and the count of "town road" is 1.
Step 4: {"sheep": 3, "woman": 2, "town road": 1}
Plausible scene: {"sheep": 3, "woman": 2, "town road": 1}
```

```
Other examples with skipped step-by-step process:

Prompt: A desk and office chair in the cubicle
Plausible scene: {"office desk": 1, "office chair": 1, "cubicle": 1}

Prompt: A pizza is in a box on a corner desk table.
Plausible scene: {"pizza": 1, "box": 1, "desk table": 1}

Note: Print **ONLY** the final scene as a JSON object.
```

## A.3 HUMAN EVALUATION

**Setup**  We randomly sample 100 captions from the COCO captioning dataset and generate layouts for each caption using LayouSyn, LayoutGPT with GPT-3.5, and LayoutGPT with GPT-4. We present the generated layouts to human raters and ask them to assign a score between 1 (strongly disagree) to 5 (strongly agree) for how well the layout represents the caption. We design an interface for human evaluation as shown in Fig. A.3. The interface displays the caption, the generated layout, and 5 radio buttons for raters to assign a score. In our current batch, we ask graduate student volunteers to rate the layouts. A total of 9 raters participated in the evaluation, resulting in a total of 1380 rated layouts and on average 4.6 ratings per layout. We plan to conduct a larger-scale evaluation on AMT in the future.

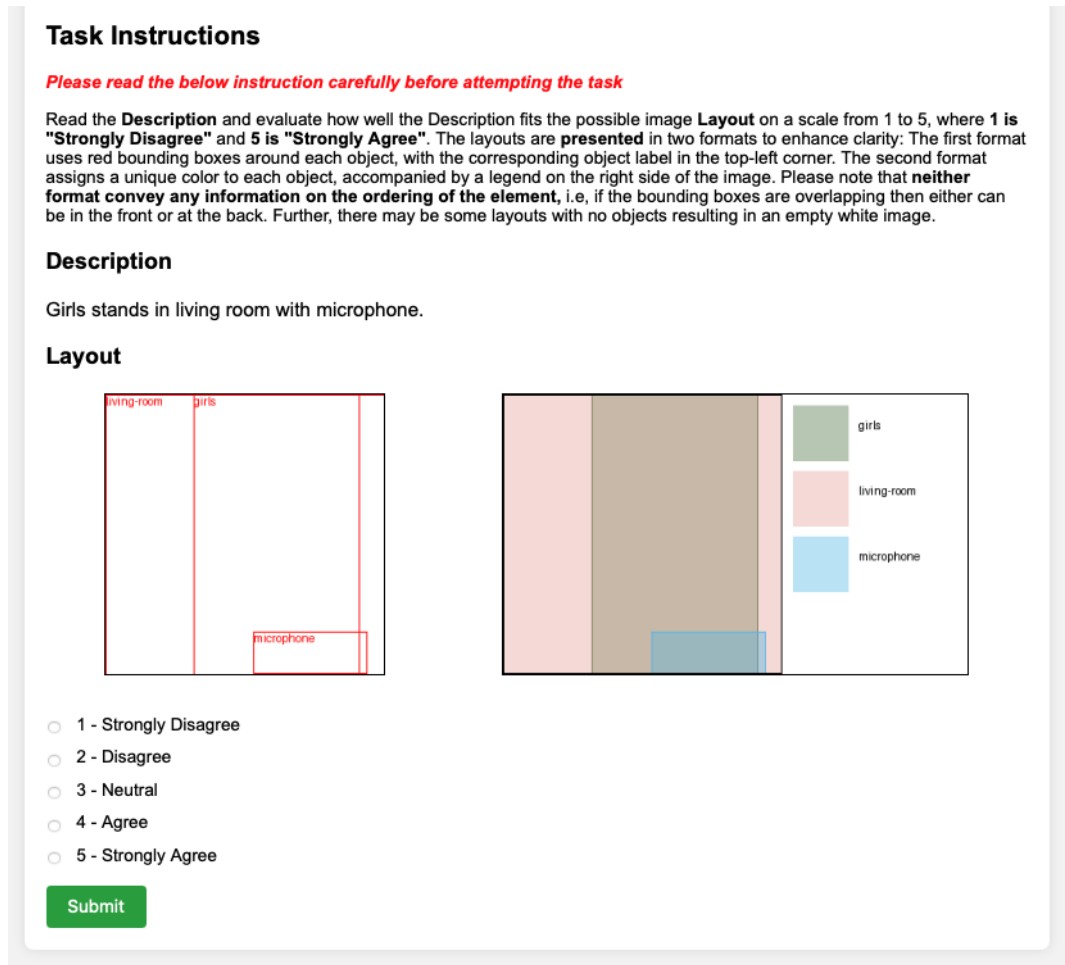

Figure A.3: **Interface for human evaluation**: Raters assign a score between 1 (strongly disagree) to 5 (strongly agree) for the quality of the layout.

