# OpenReview forum: "Lay-Your-Scene: Open-Vocabulary Text to Layout Generation"
_ICLR.cc/2025/Conference — ICLR 2025 Conference Withdrawn Submission_

### Official Review · Reviewer_neG7 · 2024-10-31

**Soundness:** 3
**Presentation:** 3
**Contribution:** 2
**Rating:** 5
**Confidence:** 4

**Summary:**

This paper proposes a simple framework for layout-guided image synthesis. First it trains

**Strengths:**

+ This paper propose a text-conditioned layout generator with diffusion models.

**Weaknesses:**

+ Concern on the image quality. The quality of the layout-based image generator used is GLIGEN, which is hard to compete with the state-of-the-art image generators like FLUX and SD3. However, layout conditioned image generation is not a trivial problem.
Most of the visual results are unsatisfying.

+ Moreover, using layout to describe the scene is not appropriate/too simplistic in most cases. It can not capture some challenges in real world, like the overlap, occlusion and poses for human and animals.

+ The text prompt is too simplistic with only quantities and object names.

+ It is very interesting to show the diversity of the layout generator.

+ I suggest that the usage of layout generator should be extended in more applications, not restricted in the image generation.

**Questions:**

+ See weakness.

---

### Official Review · Reviewer_f7vc · 2024-10-31

**Soundness:** 2
**Presentation:** 2
**Contribution:** 2
**Rating:** 3
**Confidence:** 5

**Summary:**

The paper introduces LayouSyn, a framework for open-vocabulary text-to-layout generation. This approach is designed to improve the generation of natural scene layouts by leveraging both language models and vision-based models. LayouSyn addresses the limitations of previous methods by using a two-stage process: first, extracting object labels from text using a lightweight language model, and second, predicting scene layouts with a conditional diffusion-transformer network. The framework shows versatility in applications such as LLM-initialization and object-addition, achieving state-of-the-art results across various benchmarks.

**Strengths:**

This paper has the following strengths:

* Versatility: LayouSyn demonstrates flexibility with its ability to initialize from LLM-generated layouts and perform object addition, showcasing its potential for diverse applications in graphical design and document analysis.

* Empirical Performance: The experimental results indicate that LayouSyn achieves superior performance in generating semantically and geometrically plausible scene layouts, outperforming existing methods on both closed and open-vocabulary benchmarks.

* Addressing Limitations of Prior Work: By reducing dependency on large language models, LayouSyn offers a more transparent and cost-effective solution, addressing issues of opacity, latency, and cost associated with LLMs.

**Weaknesses:**

Some key weakness points are as follows:

* Limited Theoretical analysis:
This paper is a more engineering-like work based on difussion transformer to achieve better layout generation. However, the theoretical contribution is incremental from the perspective of ICLR community.

* Complexity and Implementation:
The proposed two-stage framework, while effective, adds complexity to the layout generation process. Details regarding computational efficiency and scalability in practical applications could be further elucidated.

* Evaluation Metrics:
The paper could benefit from a more detailed discussion on the evaluation metrics used to assess the semantic and geometric plausibility of generated layouts. Understanding how these metrics correlate with human judgment would strengthen the claims of state-of-the-art performance for open-vocabulary layout generation.

* Ablation Studies:
While the paper presents extensive experiments, additional ablation studies focusing on the contributions of different components (e.g., language model size, transformer architecture) would provide deeper insights into the framework’s efficacy.

**Questions:**

* Writing and presentation are not good enough.
For example, the claim of "first end-to-end text-tonatural-scene-layout generation pipeline" is self-contained.  Meticulous reading of this paper could not even enable readers to clarify what is important for "first". Indeed,  the "first" is merely an engineering-level improvement. There are many of such inappropriate writing contents.
Also, Figures are not well drawn. For example Fig.2 has many unclear denotations, such as "T5" and "LM" and lines with unclear colors. While this is only clear to authors not to readers.

 * Concerns for open-vocabulary generation. The experiments are only done with the benchmark datasets. While this paper claims to be robust for diverse scenes, there are no results showing how can the model generalize to real-world layout generation.

Upon the weakness and questions, I am lean to reject. This paper is incremental in theoretical breakthrough and method novelty.

**Details Of Ethics Concerns:**

N.A.

---

### Official Review · Reviewer_ZY83 · 2024-11-02

**Soundness:** 1
**Presentation:** 2
**Contribution:** 1
**Rating:** 3
**Confidence:** 4

**Summary:**

This work proposes a method for generating object layouts from text descriptions, which can be used for layout-to-image generation or object addition. The generation pipeline involves a large language model (LLM), a pre-trained GroundingDINO model (only used during training), and a diffusion-based generation model. The LLM extracts objects mentioned in the text description, and GroundingDINO detects the locations of these objects, serving as training targets for the diffusion model. The diffusion model is then trained to predict the coordinates of object bounding boxes, conditioned on the text encoded by a T5 model. Experiments are conducted on the NSR-1K dataset, which contains spatial relationships between instances, and COCO-GR, a processed version of COCO augmented with GroundingDINO.

**Strengths:**

* The paper introduces a method to transform captioned datasets, such as COCO, into datasets suitable for text-to-layout generation.
* The method demonstrates potential for enabling object addition within images.
* The proposed approach supports various generation resolutions.

**Weaknesses:**

1. The evaluation metric used for the closed-set evaluation on the COCO dataset (Ln 301) appears problematic. Firstly, assessing image quality based on generated images from Layout2Im [A] seems inappropriate, as the proposed method focuses on layout generation, not layout-to-image generation. Using images generated by [A] to measure image quality does not accurately reflect the quality of the layout itself. Secondly, it is unclear why a layout-to-image method from 2019 [A] was chosen for image generation in this experiment, while more advanced generative models, such as GLIGEN, are used elsewhere in the paper (e.g., Figures 1 and 7).
2. The paper claims that the only prior work for open-vocabulary natural scene layout generation is LayoutGPT (Ln 370), which is therefore the only baseline. However, methods such as [B] can also be adapted for natural scene layouts, as demonstrated in their respective papers. In addition, text generation experiments in [B] suggest that their method could be applied to open-set scenarios if trained with the COCO-GR dataset used in this paper. The reviewer suggests the authors to have a more comprehensive survey on text-to-layout generation and add more baselines into the experiments.
3. The numerical reasoning experiments (Tab 5) may be unfair to the LayoutGPT baseline in terms of precision, recall, and accuracy metrics. Unlike the baseline, which must infer the number of objects directly from the description, the proposed method has pre-extracted object candidates from the LLM, with the trainable diffusion model only responsible for inferring object locations. This setup implies that the accuracy of identifying objects is entirely dependent on the LLM's performance. Given that the LLM is not fine-tuned, the performance improvement may stem from better prompt engineering rather than the proposed method itself.
4. Using CLIP similarity to evaluate numerical and spatial reasoning is not intuitive, and employing a layout-to-image model to convert generated layouts into images introduces additional uncontrollable variations in the evaluation process.
5. The proposed method uses 250 denoising steps (Ln 481), which raises concerns about throughput. While the LayoutGPT baseline can use online services for a fast response, the proposed approach requires first employing an LLM and then running a diffusion model with a large number of denoising steps. The reviewer wonders how the model's performance would be affected if a faster denoising scheduler, such as DDIM with 10 or 20 steps, were used. Additionally, it would be helpful to know the current throughput of the proposed method compared to the baseline.

[A] Image generation from layout

[B] LayoutDM: Transformer-based Diffusion Model for Layout Generation

**Questions:**

The reviewer believes that the evaluation in this paper is neither comprehensive nor robust enough to qualify as a publishable paper for a top conference such as ICLR. Also, the performance contribution of training a diffusion model for object location prediction is difficult to be clearly identified. As such, the reviewer does not think current version of the paper is ready to be accepted

---

### Official Review · Reviewer_VXWd · 2024-11-03

**Soundness:** 2
**Presentation:** 2
**Contribution:** 2
**Rating:** 5
**Confidence:** 4

**Summary:**

Summary: This work proposed an end-to-end natural-scene layout generation framework by (1) Firstly acquiring the open-vocabulary set by a small open-source LLM and corresponding layout (in bounding box format) from GoundingDINO as inputs; (2) Secondly, train the generation of the layout information based on the extracted inputs and conditioned on the original texts.

**Strengths:**

1. The way of acquiring the label set (by LLM) and corresponding layout bounding boxes (by GroundingDINO) seems valid. Also, this end-to-end framework trained in this way does seem bring additional robustness compared with other paradigms that only relies on LLM. (In the paper’s words, bringing in “the strong inductive bias of vision-based models”).
2. Table 5 seems to show the performance advantage of the proposed framework over other baselines.

**Weaknesses:**

1. The evaluation of Table 5. may be questionable in the sense that GLIDE is trained on COCO too. Since LayouSyn is also trained on image-text pairs from COCO, it may pose inherent bias. Although it is addressed as “add the COCO-GR training dataset to the in-context exemplars used by LayoutGPT” in the page 7, I still feel Table .5 cannot convey convincing messages as desired. You may need to bring some universal and training-free layout-to-image generation methods to conduct similar evaluations. One of the methods may be like [https://github.com/silent-chen/layout-guidance](this).
2. The emphasis of open-source LLM seems confusing and unnecessary. Like in Table 5., LayoutGPT also can have LLama3-8B-Instruc as its LLM backbone.
3. Limited Scientific Novelty: The key component, I am assuming, is bringing GroundingDINO into the pipeline. All the  key components here seem to be off-the-shelf and rarely altered, yielding in limited scientific novelty, which may not fit in the ICLR acceptance standard.
4. Human evaluation: Firstly, the advantage shown at table 4 is not significant. Secondly, using graduate students as the main evaluator group is not proper. Large scale and strict anonymous human evaluation is required. Thirdly, when I am reading the human evaluation setup in the Appendix, I do not believe this is a proper setting that is able to scientifically and conclusively offer us insignts on evaluated methods, which can make the evaluation outcome quite arbitrary.
5. At Figure 6, “Top”, “Bottom” indicators are confusing. They look like typos.

**Questions:**

My questions are based on the list of weakness above:

1. As stated in weakness point 1, could you please conduct similar evaluations as in Table. 5 but with a training-free and neutral (not COCO trained) layout-to-image generation method?

You can address other issues as listed in the list of weakness above.


I may adjust my ratings based on the rebuttal and discussion phase with the authors.

---

### Author Response · Authors · 2024-11-15
**Official Rebuttal Response by Authors**

We would like to thank all the reviewers for their efforts in reviewing our manuscript and would like to clarify some of the common concerns of the reviewers:


1. **Motivation and Novelty of our method**


Controllability of generation model is crucial for their widespread adoption and use in real-world applications. While recent works[1,2] an achieve satisfactory control over image generation, they require users to provide fine-grained conditioning inputs, such as plausible scene layouts. A text-to-layout generation framework is therefore necessary to reduce the manual, time-consuming, and costly effort of obtaining such layouts. Additionally, an open-vocabulary setting is necessary to ensure compatibility with these methods [1, 2].

Prior works[3,4,5] primarily focus on unconditional document layout generation and assume a fixed set of object categories. Another line of work[6,7] predominantly address this challenge by leveraging the reasoning capabilities of large language models (LLMs) like ChatGPT [32] for open-vocabulary generation.  Although these LLM-based approaches can generate reasonable scene layouts, they often produce unrealistic object aspect ratios or unnatural bounding box placements, particularly with longer scene descriptions [8]. Further, their reliance on proprietary LLMs for layout generation reduces transparency and incurs additional financial costs. In contrast, LayouSyn is the **first** method to address these challenges by adopting lightweight open-source language models to predict objects from text prompts and a novel open-vocabulary diffusion-Transformer based architecture trained in a scene-aware manner to generate layouts at any aspect ratio.

---

2. **Quality of visualized images**


We would like to clarify that layout-to-image generation is not the focus of this work, and the visualized images presented represent the current state-of-the-art results in conditional layout-to-image generation [1, 2]. Methods such as Flux and SD3 are designed for free-form image generation and are not capable of generating images conditioned on layouts.

---

3. **Evaluation metrics**

We have provided Layout-FID metric results in Table 3, which are directly evaluated on the generated layouts. Additionally, the precision, recall, and accuracy metrics in Table 5 are also assessed based on the generated layouts. Our rationale for including metrics on generated images is twofold: first, they help quantify the improvement in the downstream task of layout-to-image generation, which is the primary application area for our method. Second, previous works have used these metrics to demonstrate model improvements [6]. Therefore, we believe that reporting these metrics provides a more comprehensive evaluation of our method.


**References**

[1] Xudong Wang, Trevor Darrell, Sai Saketh Rambhatla, Rohit Girdhar, and Ishan Misra. Instancediffusion: Instance-level control for image generation.

[2] Li, Yuheng, et al. "Gligen: Open-set grounded text-to-image generation." Proceedings of the IEEE/CVF Conference on Computer Vision and Pattern Recognition. 2023.

[3] Naoto Inoue, Kotaro Kikuchi, Edgar Simo-Serra, Mayu Otani, and Kota Yamaguchi. Layoutdm: Discrete diffusion model for controllable layout generation. In Proceedings of the IEEE/CVF Conference on Computer Vision and Pattern Recognition, pages 10167–10176, 2023. 1, 3

[4] Yilin Wang, Zeyuan Chen, Liangjun Zhong, Zheng Ding, Zhizhou Sha, and Zhuowen Tu. In Dolfin: Diffusion Layout Transformers without Autoencoder, 2024. 1, 3, 5 869

[5] Chai, Shang, Liansheng Zhuang, and Fengying Yan. "Layoutdm: Transformer-based diffusion model for layout generation." Proceedings of the IEEE/CVF Conference on Computer Vision and Pattern Recognition. 2023.

[6] Feng, Weixi, et al. "Layoutgpt: Compositional visual planning and generation with large language models." Advances in Neural Information Processing Systems 36 (2024).

[7] Gani, Hanan, et al. "Llm blueprint: Enabling text-to-image generation with complex and detailed prompts." arXiv preprint arXiv:2310.10640 (2023).

[8] Hanan Gani, Shariq Farooq Bhat, Muzammal Naseer, Salman Khan, and Peter Wonka. Llm blueprint: Enabling text-to-image generation with complex and detailed prompts. In The Twelfth International Conference on Learning Representations, 2024.

---

### Note · Authors · 2024-11-15

I have read and agree with the venue's withdrawal policy on behalf of myself and my co-authors.